# Temporal Dynamics and Clinical Predictors of Brain Metastasis in Breast Cancer: A Two-Decade Cohort Analysis Toward Tailored CNS Screening

**DOI:** 10.3390/cancers17060946

**Published:** 2025-03-11

**Authors:** Fernando Terry, Eduardo Orrego-Gonzalez, Alejandro Enríquez-Marulanda, Niels Pacheco-Barrios, Martin Merenzon, Ricardo J. Komotar, Rafael A. Vega

**Affiliations:** 1Department of Neurosurgery, Beth Israel Deaconess Medical Center, Harvard Medical School, Boston, MA 02115, USA; 2Department of Neurosurgery, Johns Hopkins University School of Medicine, Baltimore, MD 21205, USA; 3Department of Neurological Surgery, Leonard M. Miller School of Medicine, University of Miami, Miami, FL 33146, USA

**Keywords:** breast neoplasms, brain metastases, detection of cancer, central nervous system

## Abstract

Breast cancer constitutes a prominent source of brain metastases (BMs), which significantly contribute to mortality among women. However, there are no established guidelines on when or how frequently brain imaging should be conducted in these patients. This retrospective study aimed to identify clinical factors that could predict overall survival and the time to the development of BMs over a 20-year period. We found that early detection of BMs (within 2 years after breast cancer diagnosis) was linked to better prognosis. Baseline factors such as post-menopausal status, ethnicity, and HER2+ tumor subtype were associated with a faster development of BMs, while treatments like adjuvant endocrine therapy and Palbociclib were found to delay BM onset. Our findings suggest that incorporating these clinical predictors into an algorithm could help improve survival and quality of life for breast cancer patients at risk of developing brain metastases.

## 1. Introduction

Breast cancer is the most common malignancies among women and the second leading cause of cancer-related mortality globally, following lung cancer [1,2]. It is also the second most common frequent origin of brain metastases (BMs), accounting for 5–20% of cases, after non-small-cell lung cancer (20–56% of cases) [3]. Metastasis to the central nervous system (CNS), including both BMs and leptomeningeal metastases (LMs), represents a critical clinical event in breast cancer, significantly affecting both survival and quality of life [4]. Approximately 30–50% of patients with metastatic breast cancer will develop BMs during their disease course [5,6,7].

Recent studies suggest an increasing incidence of CNS metastases in breast cancer, likely due to prolonged survival among metastatic patients, limitations of certain chemotherapies in crossing the blood–brain barrier, and more effective systemic targeted therapies (e.g., Human Epidermal Growth Factor Receptor 2 [HER2] inhibitors). Additionally, advancements diagnostic imaging have improved detection of CNS metastases [8,9,10,11]. Several risk factors for the development of BMs have been identified, including younger age, specific ethnic backgrounds, tumor characteristics (such as poor differentiation, hormone receptor-negative status, and HER2-positive subtypes), the presence of more than four metastatic lymph nodes, and certain genetic variations [12,13].

The prognosis for patients with CNS metastases remains poor, with median overall survival (OS) ranging from 4 to 30 months, depending on factors such as age, tumor biology, disease severity, time to BM development, number of BMs, and treatment approach [14,15,16]. For instance, whole-brain radiation therapy (WBRT) is associated with worse outcomes compared to stereotactic radiosurgery (SRS) [17]. In contrast, a shorter interval between breast cancer diagnosis and BM detection has been linked to better survival outcomes [18]. However, routine brain screening using computer tomography (CT) or magnetic resonance imaging (MRI) is not recommended unless clinical symptoms or physical signs of CNS involvement are present.

Despite these insights, the literature remains limited regarding clinical predictors of survival outcomes in breast cancer patients, and clear guidelines for when to initiate CNS imaging are lacking. Published data on baseline clinical characteristics that could guide clinical decision-making for CNS imaging screening in breast cancer patients are sparse. This study aimed to identify clinical predictors associated with OS and time to BM development in female breast cancer patients, with the goal of informing screening protocols and improving patient outcomes.

## 2. Materials and Methods

### 2.1. Study Cohort and Patient Selection

This retrospective analysis utilized a prospectively collected database from a single brain tumor center in the U.S., including all female patients diagnosed with breast cancer and developed BMs between 2000 and 2020. Patients were identified from institutional medical records using ICD-10 codes C50 and C79.31. Exclusion criteria included the presence of primary brain tumors mimicking BMs, any primary tumor other than breast cancer, and the absence of radiological information at the time of BM diagnosis. This study was approved by the local institutional review board (IRB), and informed consent was waived due to the observational nature of the study. The study adhered to the Strengthening Reporting of Observational Studies in Epidemiology (STROBE) reporting guidelines (Appendix A) [19].

### 2.2. Clinical, Tumor, and Treatment Characteristics

Data were collected using a standardized data sheet, including the following clinical and tumor characteristics: patient demographics (age at diagnosis, gender, ethnicity (White, African American, Asian, and others accounting for Hispanic/Arabic ethnicity), smoking history, and Karnofsky Performance Scale (KPS) score); primary tumor and BM characteristics, such as histological type (invasive ductal carcinoma, invasive lobular carcinoma, or other types), receptor type (ER+, HER2+), receptor subtype (ER+/HER2−, ER-/HER2+, ER+/HER2+, ER-/HER2−), initial tumor grade (T1-T2, T3-T4, as per the American Joint Committee on Cancer staging manual [20]), positive sentinel lymph node status, initial clinical stage (I-III, IV), and the presence of other metastases (distal lymph node, osseus, vertebral, visceral); and early BM diagnosis (<2 years after breast cancer diagnosis). Finally, we collected treatment information for the primary therapeutic approach (surgical resection, stereotactic radiosurgery (SRS), whole-brain radiotherapy (WBRT).

For tumor characteristics, HER2+ expression was defined as a strong overexpression (3+) by immunohistochemistry (ICH) or moderate (2+) overexpression with HER2 amplification (>2.0) on fluorescent in situ hybridization [21]. ER+ expression was defined as positive nuclear IHC staining > 1% [22].

### 2.3. Survival Outcomes

The primary outcome for survival analysis was OS, defined as the time from breast cancer diagnosis to death from any cause. The secondary outcome was the time to BM diagnosis, measured from the primary breast cancer diagnosis. BMs were diagnosed based on routine CT scans conducted periodically after the diagnosis of breast cancer.

### 2.4. Statistical Analysis

Categorical variables are presented as frequencies with percentages. Continuous variables are presented as median (interquartile range [IQR]) or mean ± standard deviation (SD), depending on data normality, assessed using the Shapiro–Wilk test. Survival analysis included both univariate and multivariate methods. Univariate analyses involved Kaplan–Meier curves to assess time to overall mortality (stratified by time to diagnosis of BMs < 2 years, initial therapeutic approach, and the presence of vertebral metastases) and time to BM diagnosis (stratified by tumor subtype, ethnicity, use of hormone therapy, and use of Palbociclib). Bivariate analysis was conducted using the log-rank test for comparisons between groups. For multivariate analysis, clinically significant variables identified in univariate analysis were included in the Cox Proportional Regression models. Both unadjusted and adjusted Cox regression models were built to identify significant clinical predictors of time to BM diagnosis and OS. Hazard Ratios (HRs), 95% Confidence Intervals (CIs), and *p*-values are reported. Statistical significance was set at *p* < 0.05. All statistical analyses were performed using STATA 18.0/BE.

## 3. Results

### 3.1. Baseline Characteristics

Of the 254 patients initially identified through retrospective screening of medical records, 141 were excluded (18 due to loss to follow-up post-treatment) (Figure 1). A total of 113 female patients were included in the final analysis. All of them underwent CNS screening upon identification of clinical findings. The mean age of breast cancer diagnosis was 53.5 ± 11.27 years. The majority of the cohort was older than 50 years (n = 66, 58.4%), White (n = 76, 70%), and had a prior smoking history (n = 31, 27.4%). The most common histopathological subtype was invasive ductal carcinoma (n = 82, 72.6%), with a predominant IHC profile of ER+ (n = 82, 73.2%). The most frequent receptor subtype was ER+/HER2− (n = 58, 51.8%). Among ER+ patients, the majority received hormone therapy (n = 41, 48.8%) and Palbociclib (n = 14, 16.7%).

Regarding tumor characteristics, most patients presented with T3–T4-grade tumors (n = 59, 52.2%), clinical stage I–III at diagnosis (n = 80, 70.8%), a positive sentinel lymph node (n = 20, 17.7%), and a low KPS score (n = 68, 60.2%). At the time of breast cancer diagnosis, metastases were already present in distal lymph nodes (n = 33, 29.2%), bones (n = 28, 24.8%), vertebrae (n = 25, 22.1%), and viscera (n = 44, 38.9%), before intracranial dissemination. The most common therapeutic approach for BMs was WBRT (n = 53, 46.9%). A detailed summary of baseline characteristics is presented in Table 1.

### 3.2. Survival Analysis

The median follow-up time for all patients was 12.6 years (IQR: 4.4–30.6). The median time to BM diagnosis, from breast cancer diagnosis, was 4.87 years (IQR: 2.8–11.2). Kaplan–Meier curves and log-rank tests were performed to evaluate univariate clinical predictors of OS and time to BM diagnosis. Key baseline characteristics were independently assessed within subgroups.

### 3.3. Univariate Analysis

Log-rank tests identified the following as significant predictors of earlier overall mortality: WBRT as the initial therapeutic approach (*p* = 0.07), a time to BM diagnosis longer than 2 years (*p* = 0.016), and the presence of vertebral metastasis (*p* = 0.048) (Figure 2). Similarly, for predicting a shorter time to BM diagnosis, clinically significant factors included ER+/HER2+ subtype status (*p* < 0.05), absence of hormone therapy (*p* = 0.03), and absence of Palbociclib use (*p* = 0.03) (Figure 3).

### 3.4. Cox Regression Model for Overall Survival

The first Cox regression model was constructed to identify clinical predictors for OS after BM diagnosis. The unadjusted model revealed the following significant predictors of survival: positive sentinel lymph node status (HR = 2.1; 95% CI: 1.01–4.3, *p* = 0.041), presence of vertebral metastasis (HR = 2.01; 95% CI: 1.01–4.28, *p* = 0.046), early BM diagnosis (HR = 0.24; 95% CI: 0.074–0.83, *p* = 0.025), and WBRT as the initial treatment (HR = 2.7; 95% CI: 1.11–6.5, *p* = 0.03). After adjusting for the baseline characteristics, only early BM diagnosis remained a strong predictor of prolonged OS (HR = 0.22; 95% CI: 0.049–0.98, *p* = 0.048) (Table 2).

### 3.5. Cox Regression Model for Time to BM Diagnosis

A second Cox regression model was built to determine clinical predictors for time to BM diagnosis. In the unadjusted model, the following factors were associated with a shorter time to BM diagnosis: post-menopausal state at initial breast cancer diagnosis (HR = 1.49; 95% CI:1.02–2.21, *p* = 0.04), other ethnicities (Hispanic or Arabic) (HR = 1.49; 95% CI 1.02–2.21, *p* = 0.04), HER2+ status (HR=2.19; 95% CI 1.26–3.82, *p* = 0.01), and ER-/HER2+ (HR = 2.21; 95% CI 1.22–4.01, *p* = 0.01) and ER+/HER2+ (HR=2.14; 95% CI 1.25–3.65, *p* = 0.01) subtypes. After adjusting for other baseline characteristics, post-menopausal status (HR = 1.69; 95% CI: 1.13–2.53, *p* = 0.01), other ethnicities (HR = 2.45; 95% CI: 0.98–6.07, *p* = 0.053), and ER+/HER2+ subtype (HR = 2.06; 95% CI: 1.14–3.71, *p* = 0.016) remained significantly associated with shorter time to BM diagnosis (Table 3).

### 3.6. Cox Regression Model for ER+ Subgroup

A third Cox regression model focused on ER+ breast cancer patients revealed that the ER+/HER2+ subtype (HR = 1.81, 95% CI: 1.09–2.96, *p* = 0.02) and stage IV disease at diagnosis (HR = 1.83, 95% CI: 1.1–3.18, *p* = 0.03) were significant predictors of shorter time to BM diagnosis in the unadjusted model. Conversely, patients that received adjuvant endocrine therapy (HR = 0.61; 95% CI: 0.39–0.95, = 0.03) or used Palbociclib (HR = 0.51; 95% CI: 0.28–0.96, *p* = 0.04) had a significantly longer time to BM diagnosis. In the adjusted model, both adjuvant endocrine therapy (HR = 0.65; 95% CI: 0.39–1.06, *p* = 0.09) and Palbociclib use (HR = 0.57; 95% CI: 0.29–1.10, *p* = 0.09) remained significant (Table 4).

A summary of the clinical predictors is presented in Figure 4.

## 4. Discussion

This two-decade-long retrospective cohort study provides valuable insights into the natural history of BMs in female breast cancer patients. Our findings revealed a median follow-up time of 12.6 years (IQR: 4.4–30.6) and a median time to BM diagnosis of 4.87 years (IQR: 2.8–11.2), which aligns with previously published reports on BM development in this patient population [12,18]. Notably, our Cox regression analysis identified several clinical predictors associated with a shorter time to BM development, including post-menopausal status, Hispanic or Arabic ethnicity, and the ER+/HER2+ tumor subtype. Conversely, the use of adjuvant endocrine therapy and Palbociclib was associated with delayed BM onset. Furthermore, an early diagnosis of BMs (<2 years after breast cancer diagnosis) emerged as a strong predictor for prolonged OS, while WBRT as the initial treatment was linked to poorer OS.

### 4.1. Clinical Predictors of Time to Overall Survival

The initial unadjusted Cox regression model identified several clinical features that were associated with poorer OS, such as positive sentinel lymph node status, vertebral metastases, and WBRT as first-line therapy. The association between positive sentinel nodes (defined by sentinel biopsy (SNB)) and OS aligns with our current clinical understanding, where SNB is a critical factor guiding surgical management decisions. Emerging trends suggest that the role of SNB might be reconsidered in patients with clinical node-negative breast cancer, with some advocating for omitting axillary surgery altogether [23].

In line with prior literature, vertebral metastasis was also identified as a significant mortality risk factor. Studies have shown that patients with vertebral metastases often face mortality rates comparable to those with intracranial dissemination, with OS ranging from 7 to 17 months depending on the treatment modality [24,25]. Similarly, the need for WBRT, typically an indicator for advanced intracranial disease, was associated with a shorter OS. Previous studies have reported an estimated median OS of 3 months post-WBRT, with only 8.8% survival at the third year post-diagnosis of BMs [26]. In contrast, patients with BMs treated with surgery (single symptomatic lesion) or SRS showed better survival, with an estimated OS of 5.7 years and 21 months, respectively [27,28,29,30].

Interestingly, our study found that an early BM diagnosis (within 2 years from breast cancer onset) was significantly associated with prolonged OS. This finding contradicts the conventional belief that later BM development indicates worse prognosis [18,31]. Prior studies, such as that by Hulsbergen et al., have suggested a higher mortality rate in patients diagnosed with BM more than 2 years after initial breast cancer diagnosis [18]. Our results suggest that an early BM diagnosis might reflect better treatment options or less extensive intracranial spread, potentially improving survival outcomes. This emphasizes the need to explore clinical predictors for early BM detection, which could be crucial for improving life expectancy in patients with high-stage breast cancer.

### 4.2. Clinical Predictors of Time to BM Diagnosis

Post-menopausal status was strongly associated with a shorter time to BM development. This can be explained by the hormonal shifts occurring post-menopause, particularly the increase in “free androgen index” (FAI). As ovarian estrogen levels decline, androgen secretion decreases more slowly, leading to a hormonal imbalance that may enhance cellular growth and proliferation in hormone-sensitive tissues, including the breast tissue [32]. Previous studies have shown that androgens, particularly dehydroepiandrosterone sulfate (DHEA-S), testosterone, and dihydrotestosterone (DHT), may facilitate breast cancer progression and metastasis. The androgen-receptor-mediated epithelial-to-mesenchymal transition (EMT) has been implicated in promoting metastasis in murine models, including in the brain [33].

Ethnicity, particularly Hispanic or Arabic backgrounds, is another factor linked to shorter time to BM development. The literature suggests that racial and ethnic minorities, such African American and Hispanic/Latino patients, often experience faster-growing tumors, which may be linked to delayed access to healthcare or treatment. These groups tend to present with more advanced disease at diagnosis, which likely contributes to the faster progression to metastasis [34,35]. On the other hand, it has been reported Asian women are more prone to have higher tumor grades and the HER2+ subtype compared to European white women [36]. Although this study’s sample included relatively few Hispanic and Arabic patients, our findings, which are fairly consistent with the published literature, highlight the importance of addressing healthcare disparities and the potential impact of ethnicity on tumor biology and disease progression [37].

The ER+/HER2+ subtype was significantly associated with shorter time to BM development. This finding aligns with previous research indicating that HER2+ breast cancer is highly metastatic, particularly to the CNS [38,39,40,41,42]. HER2 overexpression leads to increased tumor invasiveness proliferation [43], with evidence suggesting that HER2-HER3 heterodimers activate the PI3K–Akt signaling pathway, which plays a key role in metastasis. This pathway has been implicated in promoting the survival and growth of brain metastasis, with PI3K activation observed in 77% of brain cancer brain metastasis [44,45]. Additionally, the upregulation of immune checkpoint molecules such as PD-L1 and cytotoxic T lymphocyte-associated protein 4 (CTLA4) in the brain tumor microenvironment may facilitate immune evasion and metastasis [46,47]. A study by Avila et al. [48] reported that HER2+ breast cancer patients exhibit a higher prevalence of brain metastases throughout the course of the disease due to their persistent cumulative risk, reflecting heightened CNS tropism. This underscores the importance of periodic screening for this subgroup. In contrast, the risk of brain metastases in ER+/HER2− breast cancers tends to plateau and stabilize over time, suggesting that routine cerebral imaging could be safely reduced or even discontinued after a certain point. In such cases, clinical management may shift to prioritizing clinical symptom-based diagnosis, which has demonstrated greater sensitivity in detecting new CNS events and may be more effective in guiding follow-up care than routine imaging.

In the ER+ breast cancer subgroup, we identified that the use of adjuvant endocrine therapy and Palbociclib were protective factors against BM development. Adjuvant endocrine therapies, such as tamoxifen, selective estrogen receptor modulators (SERMs), and aromatase inhibitors (AIs), have been shown to delay metastasis, including to the CNS selective estrogen receptor modulators (SERMs), which reportedly delay metastasis to other locations, including the CNS [48,49,50,51,52,53]. Notably, tamoxifen and AIs have favorable pharmacokinetic properties, such as good CNS penetration, which may improve effectiveness in preventing brain metastases [52]. Palbociclib, a CDK4/6 inhibitor, has also demonstrated activity in preventing metastases by downregulating EMT markers such as COX-2 and vimentin, which are key drivers of metastasis [53,54]. Our results underscore the potential of targeted therapies in reducing the risk of BM development and extending survival.

### 4.3. Clinical Implications

Our findings highlight the importance of identifying clinical predictors for early BM detection, as early diagnosis is strongly associated with improved OS. Current clinical guidelines recommend staging and BM screening via fludeoxyglucose-18 (FDG) positron emission tomography/computed tomography (PET/CT) for patients in clinical stage IIB or higher, who are at greater risk for CNS metastasis [55]. The literature also supports the routine use of PET imaging to modify staging, although brain MRI is not commonly used for initial BM screening [56,57]. Notably, CNS scanning in patients with extracranial metastases of recurrent breast cancers is generally discouraged, even in molecular subtypes with known CNS tropism, such as HER2+ or triple-negative breast cancers [58,59,60]. The European Society for Medical Oncology (ESMO) guidelines recommend that screening in asymptomatic patients should only be considered if it would impact systemic treatment decisions [61].

Our study suggests that factors such as post-menopausal status, Hispanic/Arabic ethnicity, and the ER+/HER2+ tumor subtype could serve as key variables to guide the need for screening aimed at early BM detection.

For patients with these identified risk factors, early BM screening (within 2 years of breast cancer diagnosis) should be prioritized. Those with ER+/HER2+ tumors or from Hispanic/Arabic ethnic backgrounds are at higher risk of earlier BM development, often before the 100-month follow-up mark. Notably, patients who did not receive hormone therapy experienced a more rapid increase in BM diagnosis between 100 and 150 months of follow-up. These findings underscore the importance of a tailored screening approach that incorporates clinical and molecular predictors to optimize early detection and outcomes in high-risk patients. Based on these insights, we propose a potential CNS screening algorithm that targets patients based on the identified predictors of BM development (Figure 5).

### 4.4. Limitations and Strengths

This study has several limitations. First, the retrospective design introduces potential for recall bias and may exclude patients with incomplete data or those lost to follow-up. Second, the “time to BM diagnosis” is based on the date of diagnosis rather than the actual onset of BM, which could lead to inaccuracies in survival analysis. Moreover, brain imaging is typically performed only upon the appearance of clinical manifestations, resulting in a highly heterogeneous screening timeline among patients. While we partially mitigated this issue by adjusting for potential confounders in Cox regression models, some variables (such as variations in individual treatment regimens, follow-up durations, and other patient-dependent factors) were not controlled for, representing an inherent source of bias. Third, the single-center design limits the generalizability of our findings, as the population was largely homogenous, representing a higher socio-economic status with access to private healthcare. Lastly, the relatively small sample size (n = 113) limits the statistical power of our analysis. Despite these limitations, this study provides valuable insights into the clinical predictors of BM development in breast cancer patients. Larger, multi-center studies are needed to confirm these findings and refine screening strategies for early BM detection, ultimately improving patient outcomes.

## 5. Conclusions

Breast cancer remains a major global health concern, with BMs contributing significantly to morbidity and mortality. Early BM detection has been shown to improve long-term survival; however, clear guidelines for screening remain lacking. Our study identifies key clinical predictors—post-menopausal status, Hispanic or Arabic ethnicity, and the HER2+ tumor subtype—that are associated with a shorter time to BM development, offering potential targets for early screening in high-risk patients. Additionally, the use of adjuvant endocrine therapies and Palbociclib were found to delay BM onset, highlighting the role of personalized treatment in preventing CNS metastases. While promising, these findings require validation in a larger prospective and multi-centered study to inform the development of a comprehensive CNS screening algorithm for breast cancer patients. Ultimately, improving early detection strategies could enhance survival outcomes and quality of life for those at risk of BMs.

## Figures and Tables

**Figure 1 cancers-17-00946-f001:**
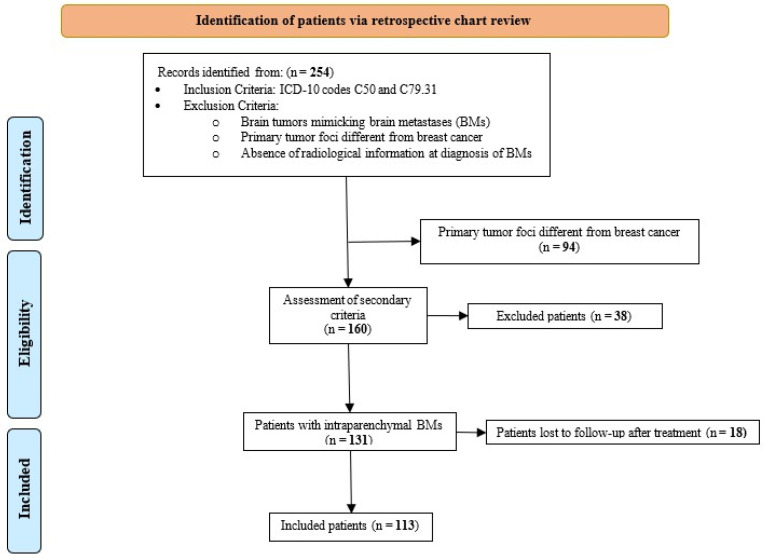
STROBE patient selection flowchart.

**Figure 2 cancers-17-00946-f002:**
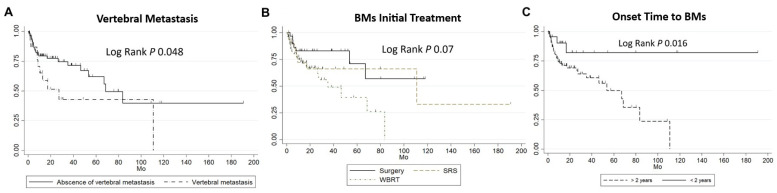
Kaplan–Meier curves reflecting time to overall mortality: (**A**) Data stratified by the presence of vertebral metastasis. (**B**) Data stratified by treatment group—surgery, stereotactic radiosurgery (SRS), and whole-brain radiotherapy (WBRT). (**C**) Data stratified by timing of BM diagnosis: early (<2 years) and late (>2 years from breast cancer diagnosis).

**Figure 3 cancers-17-00946-f003:**
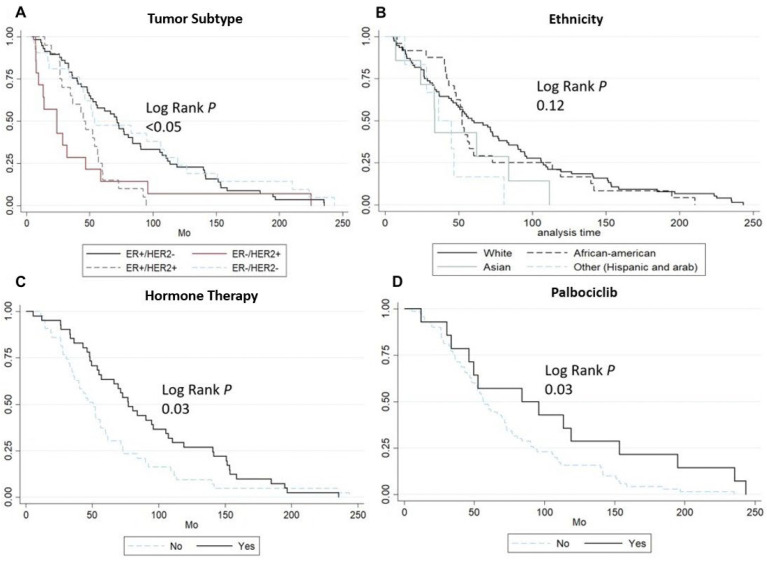
Kaplan–Meier curves reflecting time to BM diagnosis: (**A**) Data stratified by molecular receptor subtype—ER+/HER2−, ER+/HER2+, ER-/HER2+, and ER-/HER2−. (**B**) Data stratified by ethnicity—White, Asian, African American, and Other (Hispanic and Arabic). (**C**) Data based on adjuvant endocrine therapy use. (**D**) Data based on Palbociclib use.

**Figure 4 cancers-17-00946-f004:**
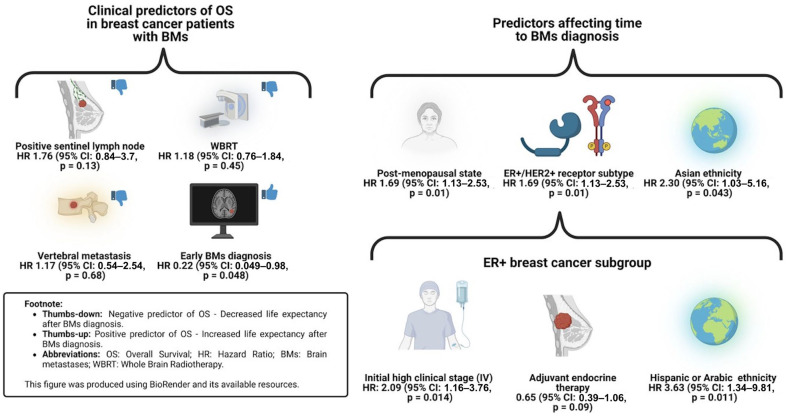
Clinical predictors of OS and time to BM diagnosis.

**Figure 5 cancers-17-00946-f005:**
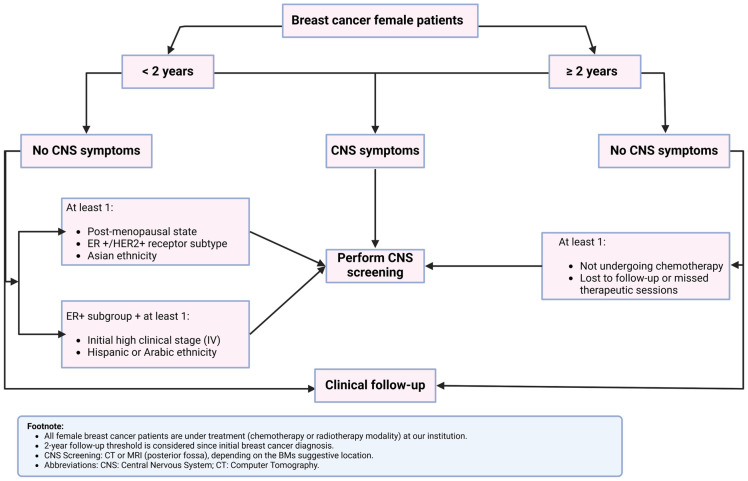
CNS imaging algorithm.

**Table 1 cancers-17-00946-t001:** Baseline demographic and clinical characteristics of the patients **.

Baseline Characteristics		Total (n = 113)
Age in years, mean ± SD		53.5 ± 11.27
Age ≥ 50 years, n (%)		66 (58.4)
Smoking history, n (%)		31 (27.4)
Ethnicity, n (%)	White	76 (70)
	African American	24 (18.5)
	Asian	8 (7.1)
	Other (i.e., Hispanic, Arabic)	5 (4.4)
Histological type, n (%)	Ductal Carcinoma	82 (72.6)
	Lobular Carcinoma	17 (15)
	Other	4 (3.5)
	Not available	10 (8.9)
Receptor type, n (%) *	ER+	82 (73.2) *
	HER2+	35 (31.3) *
Receptor subtype, n (%) *	ER+/HER2−	58 (51.8) *
	ER−/HER2+	13 (11.6) *
	ER+/HER2+	20 (17.9) *
	ER−/HER2−	21 (18.8) *
Initial tumor grade, n (%)	T1–T2	33 (29.2)
	T3–T4	59 (52.2)
	Not available	21 (18.6)
Positive sentinel lymph node, n (%)	Yes	20 (17.7)
	No	75 (66.4)
	Not available	18 (15.9)
Initial clinical stage, n (%)	I–III	80 (70.8)
	IV	24 (21.2)
	Not available	9 (8)
Distal lymph node metastasis, n (%)		33 (29.2)
Osseus metastasis, n (%)		28 (24.8)
Vertebral metastasis, n (%)		25 (22.1)
Visceral metastasis, n (%)		44 (38.9)
Low KPS at initial BM diagnosis, n (%)		68 (60.2)
BM therapeutic approach, n (%)	Open surgery	31 (27.4)
	SRS	29 (25.7)
	WBRT	53 (46.9)

* receptor type and subtype were not determined in one patient (n = 112). ** percentages may not total 100 because of rounding. Abbreviations: BMs: brain metastases; ER: estrogen receptor; HER2: Human Epidermal Growth Factor Receptor 2; SRS: stereotactic radiosurgery; WBRT: whole-brain radiation therapy.

**Table 2 cancers-17-00946-t002:** Cox regression model for clinical predictors affecting overall survival (OS) after BM diagnosis.

Clinical Predictors			Unadjusted Cox Regression	Adjusted Cox Regression
Age ≥ 50 years old at breast cancer diagnosis			1.54 (95% CI: 0.75–3.16, *p* = 0.24)	-
Age ≥ 55 years old at BM diagnosis			2.02 (95% CI: 0.91–4.44, *p* = 0.083)	-
Post-menopausal state			1.77 (95% CI: 0.89–3.5, *p* = 0.1)	-
Smoking history			1.54 (95% CI: 0.78–3.01, *p* = 0.21)	-
Ethnicity	White		1	-
	African American		0.88 (95% CI: 0.33–2.33, *p* = 0.79)	-
	Asian		1.49 (95% CI: 0.44–5.05, *p* = 0.52)	-
	Other (i.e., Hispanic or Arabic)		0.89 (95% CI: 0.14–3.86, *p* = 0.89)	-
Histological type	Invasive ductal carcinoma		1	-
	Invasive lobular carcinoma		1.72 (95% CI: 0.79–3.72, *p* = 0.17)	-
	Other		0.25 (95% CI: 0.033–1.85, *p* = 0.18)	-
Receptor type	ER+		1.66 (95% CI: 0.72–3.83, *p* = 0.23)	-
	HER2+		1.01 (95% CI: 0.49–2.04, *p* = 0.89)	-
Receptor subtype	ER+/HER2−		1	-
	ER−/HER2+		0.69 (95% CI: 0.24–2.01, *p* = 0.49)	-
	ER+/HER2+		1.58 (95% CI: 0.69–3.57, *p* = 0.27)	-
	ER−/HER2−		1.17 (95% CI: 0.41–3.29, *p* = 0.76)	-
High tumor grade (III–IV)			1 (95% CI: 0.48–2.07, *p* = 0.99)	-
Positive sentinel lymph node			2.1 (95% CI: 1.01–4.3, *p* = 0.041)	1.76 (95% CI: 0.84–3.7, *p* = 0.13)
Initial high clinical stage (IV)			1.7 (95% CI: 0.9–3.5, *p* = 0.13)	-
Distal lymph node metastasis			1.57 (95% CI: 0.78–3.17, *p* = 0.21)	-
Osseus metastasis			1.43 (95% CI: 0.72–2.84, *p* = 0.3)	-
Vertebral metastasis			2.01 (95% CI: 1.01–4.28, *p* = 0.046)	1.17 (95% CI: 0.54–2.54, *p* = 0.68)
Visceral metastasis			0.94 (95% CI: 0.47–1.87, *p* = 0.86)	-
Low KPS at initial BM diagnosis			1.64 (95% CI: 0.73–3.7, *p* = 0.23)	-
Early BM diagnosis (<2 years)			0.24 (95% CI: 0.074–0.83, *p* = 0.025)	0.22 (95% CI: 0.049–0.98, *p* = 0.048)
BM therapeutic approach		Open surgery	1	-
		SRS	1.7 (95% CI: 0.63–4.6, *p* = 0.29)	-
		WBRT	2.7 (95% CI: 1.11–6.5, *p* = 0.03)	1.18 (95% CI: 0.76–1.84, *p* = 0.45)

Unadjusted and adjusted Cox regression model estimates are reported as HRs (95% CI, *p*-value). Abbreviations: HR: Hazard Ratio, CI: Confidence Interval, ER: estrogen receptor, and HER2: Human Epidermal Growth Factor Receptor 2.

**Table 3 cancers-17-00946-t003:** Cox regression model for clinical predictors affecting time to BM diagnosis.

Clinical Predictors		Unadjusted Cox Regression	Adjusted Cox Regression
Age ≥ 50 years old at breast cancer diagnosis		1.45 (95% CI: 0.98–2.15, *p* = 0.06)	-
Post-menopausal state		1.49 (95% CI: 1.02–2.21, *p* = 0.04)	1.69 (95% CI: 1.13–2.53, *p* = 0.01)
Smoking history		1.05 (95% CI: 0.69–1.59, *p* = 0.81)	-
Ethnicity	White	1	-
	African American	1.16 (95% CI: 0.73–1.84, *p* = 0.54)	1.21 (95% CI: 0.75–1.96, *p* = 0.43)
	Asian	1.73 (95% CI: 0.79–3.79, *p* = 0.17)	2.30 (95% CI: 1.03–5.16, *p* = 0.043)
	Other (i.e., Hispanic or Arabic)	2.40 (95% CI: 1.02–5.63, *p* = 0.04)	2.45 (95% CI: 0.98–6.07, *p* = 0.053)
Histological type	Invasive Ductal Carcinoma	1	-
	Invasive Lobular Carcinoma	0.81 (95% CI: 0.49–1.34, *p* = 0.41)	-
	Other	0.93 (95% CI: 0.43–2.03, *p* = 0.18)	-
Receptor type	ER+	0.76 (95% CI: 0.49–1.16, *p* = 0.21)	-
	HER2+	2.19 (95% CI: 1.26–3.82, *p* = 0.01)	1.27 (95% CI: 0.37–4.27, *p* = 0.71)
Receptor subtype	ER+/HER2−	1	-
	ER−/HER2+	2.21 (95% CI: 1.22–4.01, *p* = 0.01)	2.39 (95% CI: 0.65–8.85, *p* = 0.19)
	ER+/HER2+	2.14 (95% CI: 1.25–3.65, *p* = 0.01)	2.06 (95% CI: 1.14–3.71, *p* = 0.016)
	ER−/HER2−	0.89 (95% CI: 0.53–1.49, *p* = 0.66)	0.99 (95% CI: 0.59–1.65, *p* = 0.97)
High tumor grade (III–IV)		0.91 (95% CI: 0.59–1.41, *p* = 0.69)	-
Positive sentinel lymph node		1.26 (95% CI: 0.76–2.1, *p* = 0.36)	-
Initial high clinical stage (IV)		1.41 (95% CI: 0.89–2.23, *p* = 0.15)	-
Distal lymph node metastasis		0.90 (95% CI: 0.59–1.37, *p* = 0.63)	-
Osseus metastasis		0.89 (95% CI: 0.58–1.38, *p* = 0.62)	-
Vertebral metastasis		0.81 (95% CI: 0.52–1.26, *p* = 0.35)	-
Visceral metastasis		0.77 (95% CI: 0.53–1.13, *p* = 0.19)	-

Unadjusted and adjusted Cox regression model estimates are reported as HRs (95% CI, *p*-value). Abbreviations: HR: Hazard Ratio, CI: Confidence Interval, ER: estrogen receptor, and HER2: Human Epidermal Growth Factor Receptor 2.

**Table 4 cancers-17-00946-t004:** Cox regression model for clinical predictors affecting time to BM diagnosis in patients with ER+ breast cancer.

Clinical Predictors		Unadjusted Cox Regression	Adjusted Cox Regression
Age ≥ 50 years old at breast cancer diagnosis		1.42 (95% CI: 0.90–2.22, *p* = 0.13)	-
Post-menopausal state		1.39 (95% CI: 0.89–2.17, *p* = 0.14)	-
Smoking history		1.08 (95% CI: 0.67–1.76, *p* = 0.75)	-
Ethnicity	White	1	-
	African American	1.23 (95% CI: 0.72–2.10, *p* = 0.46)	1.48 (95% CI: 0.81–2.71, *p* = 0.2)
	Asian	1.24 (95% CI: 0.44–3.44, *p* = 0.69)	2.39 (95% CI: 0.68–1.41, *p* = 0.17)
	Other (i.e., Hispanic or Arabic)	2.89 (95% CI: 1.21–6.92, *p* = 0.017)	3.63 (95% CI: 1.34–9.81, *p* = 0.011)
Histological type	Invasive Ductal Carcinoma	1	-
	Invasive Lobular Carcinoma	0.79 (95% CI: 0.47–1.33, *p* = 0.38)	-
	Other	0.78 (95% CI: 0.28–2.18, *p* = 0.78)	-
Receptor subtype	ER+/HER2+	1.81 (95% CI: 1.09–2.96, *p* = 0.02)	3.1 (95% CI: 0.93–10.41, *p* = 0.066)
High tumor grade (III–IV)		0.67 (95% CI: 0.39–1.12, *p* = 0.13)	-
Positive sentinel lymph node		1.22 (95% CI: 0.70–2.1, *p* = 0.48)	-
Initial high clinical stage (IV)		1.83 (95% CI: 1.1–3.18, *p* = 0.03)	2.09 (95% CI: 1.16–3.76, *p* = 0.014)
Distal lymph node metastasis		1.03 (95% CI: 0.63–1.66, *p* = 0.92)	-
Osseus metastasis		0.78 (95% CI: 0.5–1.2, *p* = 0.26)	-
Vertebral metastasis		0.84 (95% CI: 0.52–1.38, *p* = 0.49)	-
Visceral metastasis		0.75 (95% CI: 0.48–1.18, *p* = 0.22)	-
Adjuvant endocrine therapy		0.61 (95% CI: 0.39–0.95, *p* = 0.03)	0.65 (95% CI: 0.39–1.06, *p* = 0.09)
Palbociclib		0.51 (95% CI: 0.28–0.96, *p* = 0.04)	0.57 (95% CI: 0.29–1.10, *p* = 0.09)

Unadjusted and adjusted Cox regression model estimates are reported as HRs (95% CI, *p*-value). Abbreviations: HR: Hazard Ratio, CI: Confidence Interval, ER: estrogen receptor, and HER2: Human Epidermal Growth Factor Receptor 2.

## Data Availability

Additional information about statistical analysis can be provided upon request to the corresponding author.

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
