# Peer review of "Temporal Dynamics and Clinical Predictors of Brain Metastasis in Breast Cancer: A Two-Decade Cohort Analysis Toward Tailored CNS Screening"

_cancers, 2025, doi:10.3390/cancers17060946_

Round 1

Reviewer 1 Report

Comments and Suggestions for Authors

Terry et al. describe a retrospective analysis of a cohort of breast cancer patients with brain metastases.

The statistical analysis plan is robust, the paper is well-written.

However, this work lacks originality.

All of the authors' conclusions regarding the predictive factors for brain metastases in breast cancer are well-known (HER2 status, lymph node involvement, clinical stage, ethnicity).

The authors propose an algorithm for CNS follow-up based on significant factors identified in the multivariate analysis. I believe the molecular status of breast cancer and cumulative incidence should also be considered by oncologists. For example, in the original article by Avila et al. (PMID: 38453783), the authors describe a persistent cumulative risk in HER2-positive breast cancer, which should guide oncologists to perform cerebral imaging throughout the course of the disease. In contrast, ER+/HER2-negative cancers tend to reach a plateau, which could suggest discontinuing systematic cerebral imaging and shifting the focus to clinical symptoms.

Author Response

Dear Reviewer 1:

Thank you for your thoughtful and constructive comments. We truly appreciate your insight.

We acknowledge that the predictive factors for brain metastases, including HER2 status, lymph node involvement, clinical stage, and ethnicity, have been previously documented in the literature. However, we believe that our contribution lies in integrating these factors into a predictive algorithm that can guide clinical decision-making for CNS follow-up.

In response to your suggestion regarding the molecular status of breast cancer and its impact on cerebral imaging, we have added the following discussion to highlight this point: (Page 13, Lines 292-301)

“Consistently, a study by Avila et al48 reported that HER2+ breast cancer patients exhibit a higher prevalence of brain metastases throughout the course of the disease due to their persistent cumulative risk, reflecting heightened CNS tropism. This underscores the importance of periodic screening for this subgroup. In contrast, the risk of brain metastases in ER+/HER2- breast cancers tend to plateau and stabilize over time, suggesting that routine cerebral imaging could be safely reduced or even discontinued after a certain point. In such cases, clinical management may shift to prioritizing clinical symptom-based diagnosis, which has demonstrated greater sensitivity in detecting new CNS events and may be more effective in guiding follow-up care than routine imaging.

We believe that incorporating this information will strengthen the manuscript by offering more nuanced recommendations for CNS follow-up, particularly for patients with HER2-positive or ER+/HER2-negative subtypes.

Thank you again for your insightful input. We hope these revisions improve the manuscript and clarify our contributions to the field.

Reviewer 2 Report

Comments and Suggestions for Authors

The study " Temporal Dynamics and Clinical Predictors of Brain Metastasis  in Breast Cancer: A Two-Decade Cohort Analysis Toward Tailored CNS Screening", by Terry, et al makes a valuable contribution by identifying clinical factors that may predict the timing of brain metastases and overall survival in breast cancer patients. Its strengths lie in the focused research question on breast cancer brain metastasis, long-term data with 20 years, and rigorous statistical analysis. However, the retrospective design, limited sample size, and single-institution nature of the study suggest that further research—preferably prospective and multi-centered—is needed to validate these findings and inform screening guidelines more robustly. 

Since routine brain imaging for BMs is not standard practice and is typically performed only when symptoms arise, there may be variability in how and when patients were screened. This could affect the timing of BM detection and the associated survival outcomes. The study may have unmeasured or uncontrolled confounding factors (such as variations in treatment regimens, differences in follow-up durations, or other patient-specific variables) that could influence the observed associations.

Author Response

Dear Reviewer 2,

Thank you for your insightful feedback and for highlighting important limitations related to our study design. We acknowledge that the retrospective nature of our study and the limited sample size, restricted to a single institution, constrain the generalizability of our findings. Additionally, the inability to include more diverse variables, as you pointed out, is a limitation that we fully recognize.

To address your concerns, we have updated the ‘Limitations’ section of the manuscript as follows: (Page 14, Lines 346-352)

Moreover, brain imaging is typically performed only upon the appearance of clinical manifestations, resulting in a highly heterogeneous screening timeline among patients. While we partially mitigated this issue by adjusting for potential confounders in Cox regression models, some variables—such as variations in individual treatment regimens, follow-up durations, and other patient-dependent factors—were not controlled for, representing an inherent source of bias.”

We have also updated the ‘Conclusion’ section with your points on a large study to validate our findings: (Page 15, Line 367-370)

“While promising, these findings require validation in a larger prospective and multi-centered study to inform the development of a comprehensive CNS screening algorithm for breast cancer patients.”

We appreciate your understanding of these limitations and your recognition of their potential impact on our study’s findings. We believe this acknowledgment in the revised manuscript will help to contextualize the results and provide clarity on the limitations of the analysis.

Thank you once again for your thoughtful comments.

Reviewer 3 Report

Comments and Suggestions for Authors

The authors provide an excellent manuscript looking at a cohort of their patients with metastases from breast cancer and the factors associated with earlier diagnosis of the metastases and better overall survival. The report is well written and scientifically sound, and the findings are meaningful and impactful upon clinical care. The authors should investigate what prompted patients on study to receive cranial imaging and how these clinical signs effected the diagnosis of brain metastases in regard to timing of diagnosis. Were cases missed that had clinical findings or were there scans that were done in asymptomatic patients for surveillance? The authors support the notion that these metastases need to be found early but besides routine screening for everyone, how can they assimilate the current standard of care into their recommendations. Including this in the report will make it stronger. 

Author Response

Dear Reviewer 3,

Thank you for your thoughtful comments and helpful suggestions. Please find below our responses to your questions:

  • CNS Screening Protocol: All CNS screening in our cohort was performed following the identification of clinical findings. Routine surveillance for brain metastases in asymptomatic patients was not conducted. (Page 4, Lines 143-144)
  • Additional Information on Screening Practices: To address your point regarding the integration of current standard-of-care practices, we have included the following clarification in the manuscript: (Page 13, Line 320-326)
    “Literature also supports the routine use of PET imaging to modify staging, although brain MRI is not commonly used for initial BM screening 56,57. Notably, CNS scanning in patients with extracranial metastases of recurrent breast cancers is generally discouraged, even in molecular subtypes with known CNS tropism, such as HER2+ or triple-negative breast cancers 58–60. The European Society for Medical Oncology (ESMO) guidelines recommend that screening in asymptomatic patients should only be considered if it would impact systemic treatment decisions 61.

We believe this addition strengthens the manuscript by providing context regarding current clinical practices and screening recommendations.

Once again, we appreciate your constructive feedback and believe that these revisions will improve the manuscript's clarity and relevance to clinical practice.
